



# Fluid Models Capturing Farley–Buneman Instabilities

Enrique Rojas[1], Keaton Burns[2], and David Hysell[1]

[1]Earth and Atmospheric Sciences, Cornell University
[2]Mathematics, Massachusetts Institute of Technology

**Correspondence:** Enrique Rojas (elr96@cornell.edu)

**Abstract.** It is generally accepted that modeling Farley–Buneman instabilities require resolving ion Landau damping to reproduce experimentally observed features. Particle–in–cell (PIC) simulations have been able to reproduce most of these, but at a computational cost that severely affects their scalability. This limitation hinders the study of non–local phenomena that require three dimensions or coupling with larger–scale processes. We argue that a form of the five-moment fluid system can recreate several qualitative aspects of Farley–Buneman dynamics such as density and phase speed saturation, wave turning, and heating. Unexpectedly, these features are still reproduced even without using artificial viscosity to capture Landau damping. Comparing the proposed fluid models and a PIC implementation shows good qualitative agreement.

## 1 Introduction

Magnetized Hall-drifting electrons in the E–region ionosphere induce polarization drifts on the unmagnetized ions which tend to overshoot electrostatic equilibrium and accumulate in the crests of the local density irregularities faster than diffusion opposes them (Sahr and Fejer, 1996). This mechanism, which results in the amplitude enhancement of local perturbations, is the Farley–Buneman instability. This phenomenon has been shown to modify the mean state of the ionosphere in various ways as well as the magnetosphere morphology by modifying the local conductivity through anomalous heating and nonlinear currents (Wiltberger et al., 2017).

Linear fluid theory of Farley–Buneman instabilities predicts some aspects of the dynamics reasonably well. Furthermore, linear kinetic theory shows that ion Landau damping effectively suppresses the growth of smaller wavelengths (Schmidt and Gary, 1973), which motivates the necessity of resolving kinetic ion effects. Nevertheless, the linear theory fails to explain several features observed in the experimental data obtained with rockets and coherent backscatter radars (Oppenheim et al., 1996; Sahr and Fejer, 1996). Although PIC simulations have been able to model many aspects of the nonlinear physics of Farley–Buneman instabilities (Oppenheim et al., 2008; Oppenheim, M. and Dimant, 2013; Young, M.A. et al., 2020), their application to non–local scales has been very challenging due to the computational cost. This limitation has motivated the exploration of more cost-effective approaches like hybrid and fluid models, which often require much less computational resources because they do not resolve the velocity distribution of the plasma. For instance, Newman and Ott (1981) and Hassan et al. (2015) proposed a fully fluid dynamical system that models Landau damping with an artificial viscosity term on the momentum equation that damps large wavenumbers. Because of the isothermal approximation, these simulations could not capture wave turning and other thermal effects (Dimant and Oppenheim, 2004).




This work describes a numerical framework based on the five-moment fluid model to simulate Farley–Buneman instabilities and assesses its capability for capturing nonlinear features reproduced previously with PIC simulations. First, we used the artificial viscosity proposed by Hassan et al. (2015), including thermal effects. The simulation parameters were similar to the ones used by Oppenheim et al. (2008) to compare our results with the PIC estimates. Then, we show that most characteristic nonlinear features can be reproduced even after removing the artificial viscosity term from the five-moment system. These results are obtained even though the standard linear theory predicts that smaller structures will grow faster. Furthermore, we argue that the nonlinear signatures obtained without the artificial viscosity are more similar to the correspondent PIC estimates. This last result suggests that the proposed fluid framework may dramatically increase the scalability of Farley–Buneman simulations.

## 2   A Numerical Framework For Farley–Buneman Instabilities Based on Fluid Equations

The Farley–Buneman instability is an electrostatic process with the dominant dynamics mostly restricted to a 2D plane perpendicular to $B$. Both the magnetized electrons and unmagnetized ions collide predominantly with neutral particles (Rojas and Hysell, 2021). Therefore, assuming both species are locally Maxwellian, the following five–moment fluid system should be capable of capturing most of the important physics (Schunk and Nagy, 2009):

$$\frac{\partial n_s}{\partial t} + \nabla \cdot (n_s \boldsymbol{v}_s) = 0 \tag{1}$$

$$\frac{\partial \boldsymbol{v}_s}{\partial t} + \boldsymbol{v}_s \cdot \nabla \boldsymbol{v}_s = \frac{q_s}{m_s}(\boldsymbol{E} + \boldsymbol{v}_s \times \boldsymbol{B}) - \frac{kT_s}{m_s}\nabla \ln n_s - \frac{k}{m_s}\nabla T_s - \nu_{sn}\boldsymbol{v}_s + R(T_i, \boldsymbol{v}_i) \tag{2}$$

$$\frac{\partial T_s}{\partial t} + \boldsymbol{v}_s \cdot \nabla T_s + \frac{2}{3}T_s \nabla \cdot \boldsymbol{v}_s = \frac{2\mu_{sn}}{3k}\nu_{sn}v_s^2 - \delta_{sn}\nu_{sn}(T_s - T_n) \tag{3}$$

$$\nabla^2 \phi = -\frac{1}{\varepsilon_0}\sum_s q_s n_s \tag{4}$$

As usual, $n_s$, $\boldsymbol{v}_s$, $T_s$, $m_s$, and $q_s$ correspond to the density, velocity, temperature, mass, and charge of specie $s$, respectively. The collision frequency between specie $s$ and neutral particles is represented by $\nu_{sn}$, and $\mu_{sn}$ is the reduced mass of specie $s$ and the dominant neutral specie. The frame of reference is moving with the neutral particles at a temperature $T_n$. On the right side of equation (3), the term on the left corresponds to collisional heating, while the one on the right captures collisional cooling. The term $\delta_{sn}$ captures the fraction of energy lost by particle $s$ when it collides with a neutral (Dimant and Oppenheim, 2004). The electrostatic field was calculated by solving Poisson's equation (4).

The $R(T_i, \boldsymbol{v}_i)$ term on the right–hand side of equation (2) is a general regularization operator, and its purpose is to dampen the growth of larger wave numbers. Hassan et al. (2015) proposed a regularization term based on the ion viscosity operator but using the ion–neutral collision frequency instead of the Coulomb collision frequency. This term is used to damp large $k$-modes and is only necessary for ion scales (Rojas et al., 2016). If we denote the stress tensor by $\Pi$, this regularizing viscosity has the form $R(T_i, \boldsymbol{v}_i) = \nabla \cdot \Pi$, where:

$$\Pi = -\frac{n_i T_i}{\nu_{in}}\left(\nabla \boldsymbol{v}_i + (\nabla \boldsymbol{v}_i)^T - \frac{2}{3}\nabla \cdot \boldsymbol{v}_i I\right) \tag{5}$$



We used the operator proposed by Hassan et al. (2015) to model $R(T_i, \boldsymbol{v}_i)$ because it was successful in capturing several features of Farley–Buneman irregularities. Furthermore, the accuracy of this proxy will be assessed not by the quantitative estimates of the simulation but by whether it improves the resemblance to PIC simulations.

We chose a spectral solver to solve the five–moment system. Spectral methods are well known to have outstanding accuracy and to scale very efficiently when periodic boundaries are applicable, and no shocks or discontinuities are expected (Hesthaven et al., 2007). These criteria are satisfied in the case of Farley–Buneman irregularities. We build the numerical solver using the DEDALUS computational framework for solving general partial differential equations using sparse spectral methods (Burns et al., 2020). Our solver uses Fourier spatial discretizations with implicit integration of linear terms. The nonlinear terms are

integrated explicitly with 3/2–padding for dealiasing. This solver was comprehensively described by Burns et al. (2020).

## 3  Simulation Setup and Results

Some further simplifications can be applied to the five-moment system (1-4). We omitted the gyro motion term from the ion momentum equation for the ions and used $\mu_{in} \approx m_i/2$ and $\delta_{in} = 1$. For the electrons, $\mu_{en} \approx m_e$, no regularization is included, and $\delta_{en} = 3.5 \times 10^4$ (Dimant and Oppenheim, 2004). Furthermore, we used the simulation parameters shown in Table 1. These

parameters are the same as the ones used by Oppenheim et al. (2008) for their baseline simulation, except for the grid and box sizes, for which we used eight times fewer grid points and half the box size, respectively. Notice that the electron mass $m_e$ and the electron–neutral collision frequency $\nu_{en}$ have been artificially increased and reduced, respectively. Increasing the electron mass allows the use of larger time steps, but $\nu_{en}$ has to be reduced to maintain the same magnetization levels.

**Table 1.** Simulation parameters

| Parameter | Value |
| --- | --- |
| Grid size | $128 \times 128$ |
| Box size | 20 m |
| $\boldsymbol{B}$ | $0.5 \times 10^{-4}$ T |
| $\boldsymbol{E}_0$ | 50 mV/m |
| $n_0$ | $1 \times 10^9 \, m^{-3}$ |
| $m_i$ | $5 \times 10^{-26}$ kg |
| $m_e$ | $4 \times 10^{-29}$ kg |
| $\nu_{en}$ | $1.675 \times 10^3 \, s^{-1}$ |
| $\nu_{in}$ | $2.7 \times 10^3 \, s^{-1}$ |
| $T_0$ | 300 K |
| $dt$ | $10^{-6} s$ |

The simulation box size is several times smaller than the largest ones used for recent purely kinetic (Oppenheim et al.,

2008) and hybrid (Young, M.A. et al., 2017, 2019) Farley–Buneman simulations. In this work, we will use small simulation




boxes because we assume that most of the nonlinear features of the PIC simulations are independent of the dimensions of the simulation plane. Furthermore, our claim about an improvement in scalability is based on the assumption that fluid models of few moments scale better than PIC kinetic simulations for the same accuracy when kinetic effects are not dominant.

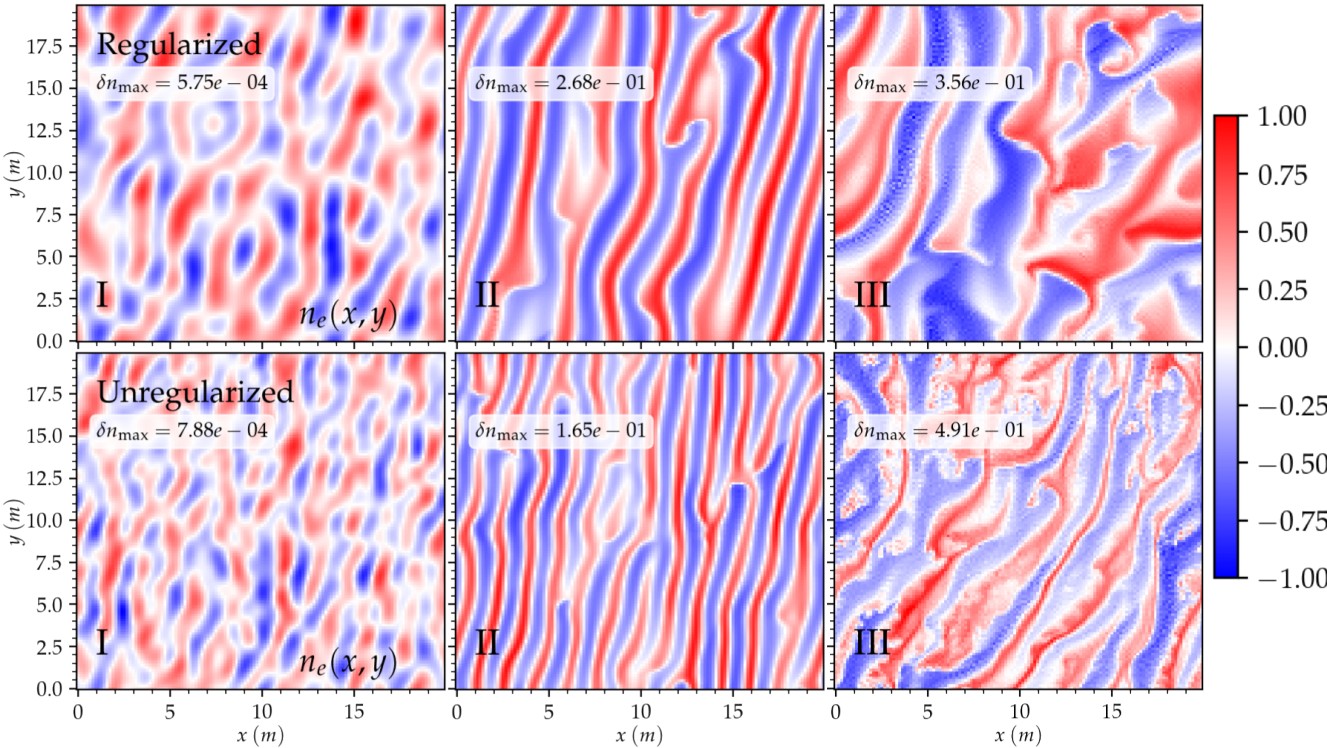

**Figure 1.** Electron density perturbation snapshot for both models. Here, $\delta n_{\max}$ indicates the maximum perturbation for that time instant. The colorbar maps the corresponding fraction of $\delta n_{\max}$. The top and bottom rows correspond to the regularized and unregularized models, respectively.

We implemented two models. We will refer to the first as "regularized" and the second as "unregularized." Both models will
solve the continuity, momentum, energy, and Poisson equations, as shown in (1-4). The only difference is that the regularized model includes the regularization operator described in equation (5), and the unregularized does not.

Figure 1 shows the electron density perturbation $n_e/n_0 - 1$ for both the regularized and the unregularized systems at representative times. Several features are common to both models. At the linear regime ($I$), we see dominant wave modes growing distinctively faster. In the mixing regime ($II$), most wave growth is aligned close to the $E \times B$ direction, and perpendicular
secondary waves start to form. After saturation, in the turbulent regime ($III$), we see a stable evolution of the electron density perturbation at around twenty percent of the background density and density structures similar to the ones obtained with PIC simulations. Moreover, we see a slight turning of the waves in the direction consistent with linear theory (Dimant and Oppenheim, 2004). On the other hand, the dominant wavelengths for the unregularized system are smaller ($\approx 1.5$ m, similar to the



PIC simulation) than for the regularized case ($\approx 2.5$ m). This difference is consistent with the idea that the regularization term

not only damps larger wave numbers but affects the dynamics of all the wave modes.

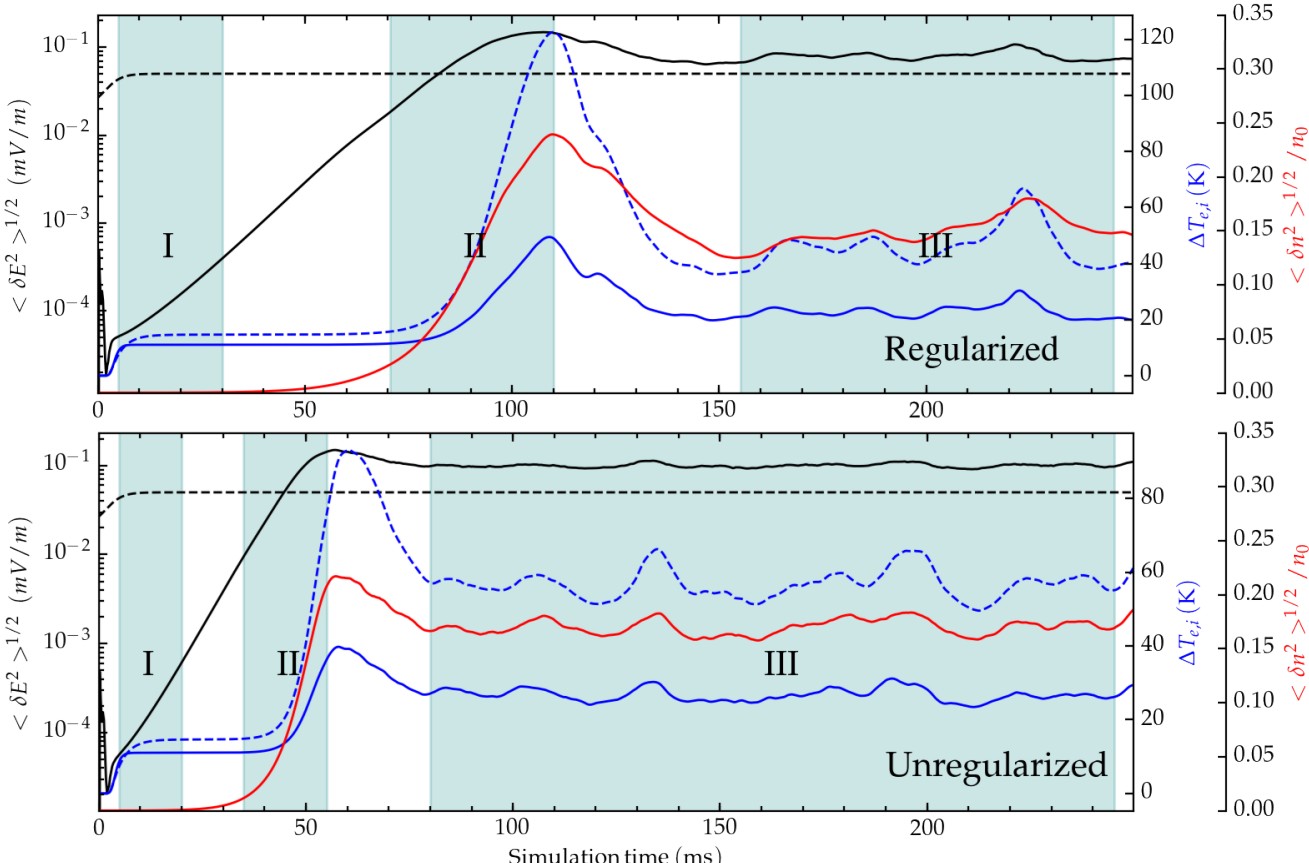

**Figure 2.** Time series for the electron temperature (blue dashed), ion temperature (blue), electron density averaged perturbation (red), background forcing field (black dashed), and root mean square of the electric field (black) for the regularized (top) and unregularized (bottom) models.

Although linear fluid theory predicts that smaller wavelengths will grow faster and destabilize the system without some form of regularization, we can see that the system not only remains stable but can capture several aspects of the expected nonlinear dynamics. This "self–regularization" may be a combination of several factors of numeric and physical origin. Even though spectral methods are well–known for having a minimal numerical diffusion compared to other approaches, this may play a

minor role in damping the larger wave numbers. Other physical mechanisms captured in this model but not in some versions of the standard linear theory and play a role in dampening smaller wavelengths are the stabilizing effect of a weak non–quasi–neutrality (Dimant, Y.S. and Oppenheim, 2011) and electron inertia (Hassan et al., 2015). An investigation of the extent of each of these mechanisms is beyond the scope of this paper and its topic of future work. Although the present simulations include




the stabilizing effect of thermal effects, we have seen similar behavior using an isothermal system, namely, the presence of a
dominant mode with no evidence of a growth rate proportional to $k^2$.

Standard metrics to diagnose the simulation, such as the root mean square (RMS) of electron density, electric field pertur-
bation, and the average heating, are presented in Figure 2. The background electric field was raised from a value below the
instability threshold at $t = 0$ to the value shown in Table 1 in the first 2 ms of the simulation. Both simulations show an increase
in temperature due to Pedersen heating in the linear regime indicated as region (I). All metrics reach saturation in the mixing
regime characterized by region (II), then stabilize in the region (III). The time series of both the regularized and, to a lesser
extent, the unregularized simulations present an overshoot just before saturation. This behavior has been documented in hybrid
(Rojas and Hysell, 2021) and fluid (Hassan et al., 2015) simulations, but it does not seem to be present in PIC simulations.

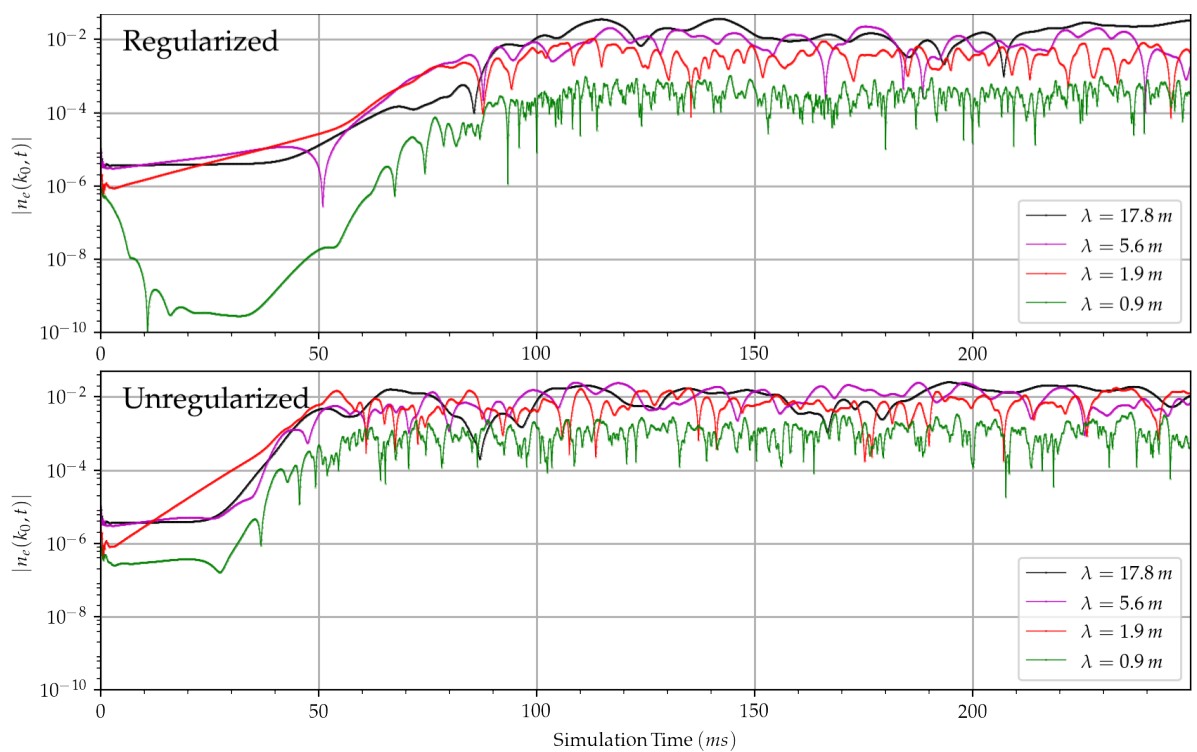

**Figure 3.** Time series of electron density perturbation spectral amplitude for specific wave numbers for both the regularized (top) and
unregularized (bottom) systems.

In both simulations, the electron density and the perturbation electric RMS field are larger than the corresponding PIC values.
The fact that this difference is more considerable for the unregularized case (around twice the PIC metrics) suggests that the
origin might be a lack of a proper damping mechanism. Although Oppenheim et al. (2008) also recovered a perturbation electric
RMS field larger than the background, the difference was smaller. They attributed this excess to factors like the truncation of
smaller wavelengths (finite box size).





For instance, the RMS field in physical space is the same as in Fourier space, so decreasing the amplitude of larger wave numbers would reduce the total RMS. Moreover, the average ion temperature evolves similarly to kinetic simulations in both cases. Nevertheless, the mean electron heating is substantially lower, probably related to the simple temperature–independent collisional heating model used in both simulations. Even though there are several quantitative differences between these and the PIC results, it is interesting to notice that in the unregularized case, the saturation onset time is much closer to the corresponding time in the kinetic simulation ($\approx 60$ ms).

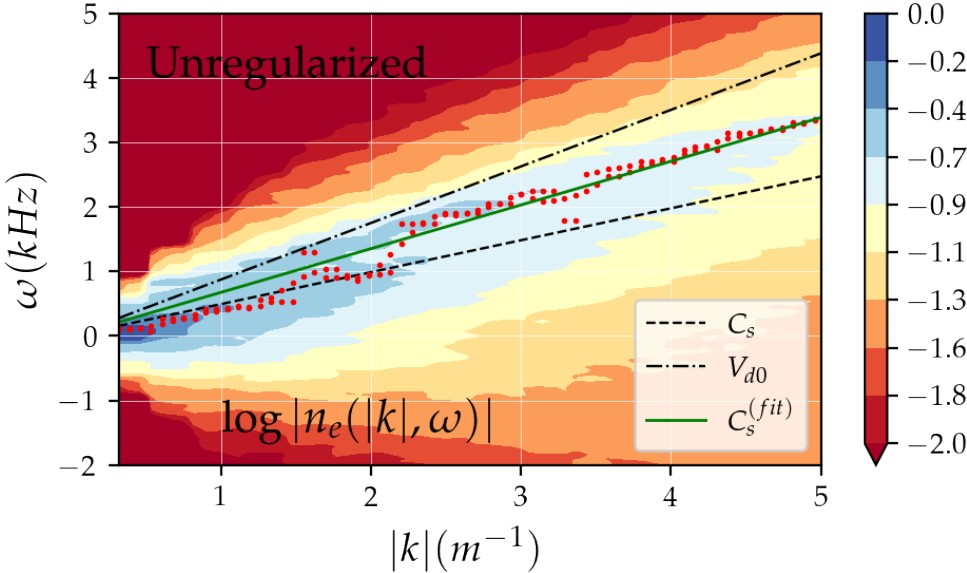

**Figure 4.** Electron density perturbation spectra for versus wave numbers for the unregularized system. The red dots indicate the maximum amplitude for a particular $\boldsymbol{k}$ and the green line represents the best–fit of $kC_s^{(fit)}$ using anomalous $\gamma_{e,i}$ as fitting parameters. $V_{d0}$ is the linear estimate of the phase speed using state parameters at the initialization.

Figure 3 illustrates the time series for the electron density perturbation for different wavelengths. Even though the regularized system shows strong damping for smaller wavelengths, both approaches seem to oscillate around similar amplitudes after saturation. This similarity may suggest that capturing the correct damping physics will affect the saturation onset time and the dominant wavelength value. Moreover, the amplitude of the larger wavelengths is comparatively larger for the regularized case.

The spectral properties of the region (III) are summarized in Figure 4 for the unregularized simulation. The spectral signatures of the regularized case are very similar but with smoother contours. The dominant wave modes propagate predominantly at phase speeds slightly above $C_s$. Heat capacity ratios representative of region (III) for both species can be estimated by fitting the expression for the ion–acoustic speed:

$$C_s^{(fit)} = \sqrt{\frac{\gamma_e \langle T_e \rangle_{\mathrm{III}} + \gamma_i \langle T_i \rangle_{\mathrm{III}}}{m_i}} \qquad (6)$$





to the most representative wave modes indicated by the red dots in Figure (4). Here, $\langle T_s \rangle_{\mathrm{III}}$ indicate the average temperature of specie $s$ over the region (III). The ion's heat capacity ratio that produced the best fit with the simulated spectral peaks was $\gamma_i \approx 2.62$ in both models, which was smaller than the one reported in the corresponding PIC simulations by 14%. On the other hand, the corresponding electron ratios were $\gamma_e^{(r)} \approx 2.32$ and $\gamma_e^{(n)} \approx 2.29$ for the regularized and unregularized cases, respectively. The obtained ratios for the electrons were larger than the obtained from PIC simulations by approximately 30%. Nevertheless, this discrepancy was expected considering the oversimplification of the constant heating and cooling rates used for both species. This simplification is especially limiting for the electron thermal evolution, considering that electron heating associated with Farley–Buneman irregularities has been observed multiple times (Bahcivan, 2007).

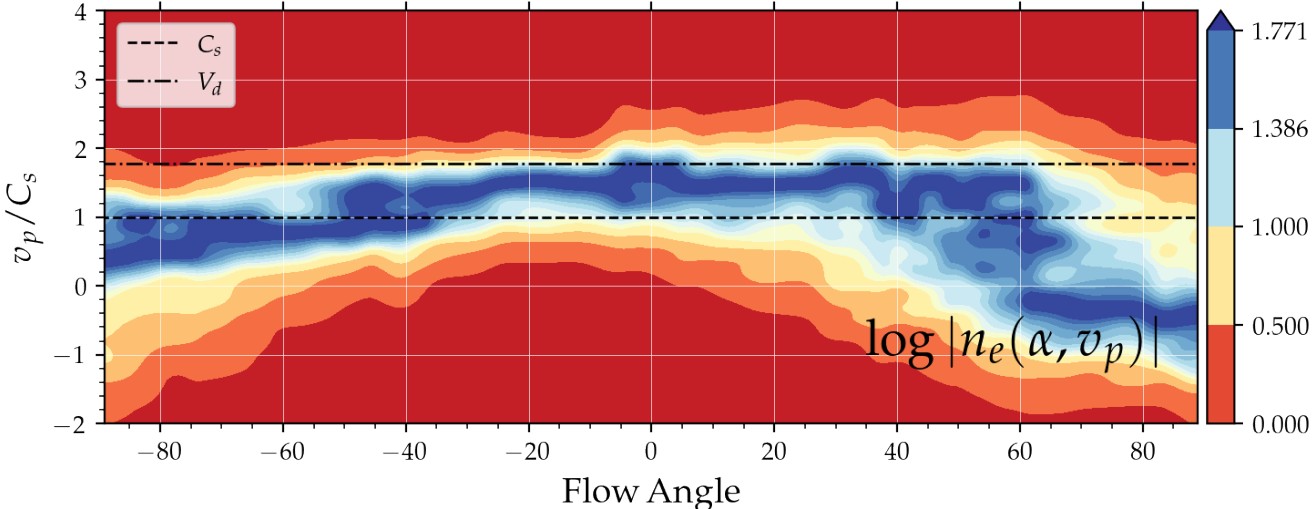

**Figure 5.** Spectra of electron density perturbations for $|\boldsymbol{k}| = 3\,m^{-1}$ with respect to phase speed and flow angle $\alpha$ for the unregularized system in region (III). The spectra was normalized with the maximum power for each flow angle. $V_d$ follows the definition of Figure **??**

The normalized spectra for $|\boldsymbol{k}| = 3\,m^{-1}$ is plotted in Figure 5 with respect to the elevation angle and the phase speed. Notice that the dominant modes have narrower widths and lie between the ion–acoustic and the convection speed. Both the Doppler shifts and the spectral widths can be calculated for each flow angle from these profiles. Considering that the system was assumed to be periodic, we can estimate the Doppler shifts and widths for the complete $360°$ by rotating the simulated ones $(d_{s,w}^{(sim)})$ appropriately. The values estimated by this rotation are labeled as $d_{s,w}^{(ext)}$. These spectral data were fitted to a modified version of a Doppler convection model used in Rojas et al. (2018):

$$d_s = \left( a + \frac{V_d^2}{b} \right) \cos(\theta + \theta_0) \tag{7}$$

$$d_w = \alpha \left( a + \frac{V_d^2}{b} \right) \sin(2(\theta - 2\theta_0)) + \beta \tag{8}$$





This empirical model relates Doppler shifts and widths to the flow angles $\theta$. The local convection velocity $V_d$, wave turning angle $\theta_0$, $a$, and $b$ are fitting parameters. The red and blue lines in Figure 6 correspond to the values of the model after

estimating the parameters using a nonlinear optimization routine. We see that this empirical model agrees closely with the simulated values. Moreover, the fitted parameters are consistent with the experimental measurements by Nielsen and Schlegel (1985).

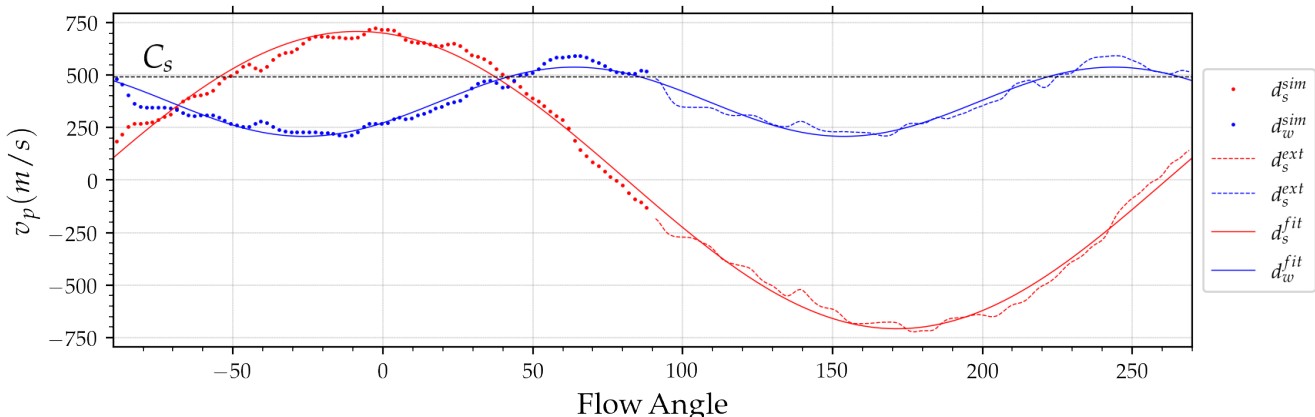

**Figure 6.** Simulated ($d_{s,w}^{(sim)}$), extended ($d_{s,w}^{(ext)}$), and fitted ($d_{s,w}^{(fit)}$) Doppler shift and spectral width calculated for regime (III) of the unregularized system. The dotted line indicates the local ion acoustic speed.

## 4   Conclusions

The results obtained with the proposed fluid models are consistent with the ones from PIC simulations in reproducing quali-

tative aspects of the Farley–Buneman instabilities. Moreover, using an unregularized five-moment fluid system, we could still reproduce most of the qualitative aspects of the diagnostics obtained with PIC simulations despite the predictions of standard linear theory. To our knowledge, this is the first time a fully fluid model is able to achieve this. Furthermore, the stabilization mechanisms of the proposed fluid electrostatic model seem to avoid the growth rates $\gamma \propto k^2$. These results suggest that we may have to reconsider the necessity for capturing ion Landau damping accurately.

Even though the simulation box sizes used in this work are small compared to the dimensions used in recent PIC simulations, the potential of scalability relies on the fact that if kinetic effects are not dominant, fluid simulations usually require much less computational resources than PIC implementations to achieve similar accuracy.

Several interesting questions were raised: What nonlinear damping processes modulate the dominant wavelengths and onset saturation time? Would it be possible to build a Landau fluid proxy that captures enough of the physics we have access to by

coherent backscatter radars? Would it be possible to modify the proposed fluid system to improve its scalability? Could these results be limited to spectral solvers? We will try to answer these questions in future studies.



Furthermore, we think this has significant implications for the plasma and space physics communities because it may open the door to other researchers having fluid plasma models to explore this topic.

*Author contributions.* ER proposed the mathematical approach for the simulation, did the numerical experiments, and wrote the manuscript. 160 KB contributed with the implementation and testing of the numerical model. DH helped designing the diagnostics to characterize the irregularities. KB and DH read the manuscript and provided feedback.

*Competing interests.* The contact author has declared that none of the authors has any competing interests.

*Acknowledgements.* This work was supported by awards AGS-1634014 and AGS-1818216 from the National Science Foundation to Cornell University.



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
