# Peer review of "Fluid Models Capturing Farley–Buneman Instabilities"

_EGUsphere, 2022_

## Author Response (AR1)

**Fluid Models Capturing Farley–Buneman Instabilities**

Response to Reviewer 1

**March 22, 2023**

Enrique Rojas1, K.J. Burns2, D.L. Hysell1 elr96@cornell.edu

1Department of Earth and Atmospheric Sciences, Cornell University 2Department of Mathematics, Massachusetts Institute of Technology

**Contents**

| 1 | General Comments to Reviewer 1 |
|---|--------------------------------|
| 2 | Response to Reviewer 1         |
|   | Comment 1                      |
|   | Comment 2                      |
|   | Comment 3                      |
|   | Comment 4                      |
|   | Comment 5                      |
|   | Comment 6                      |

**1** General Comments to Reviewer **1**

First, we want to thank you for your time writing your thorough review. Your comments were very valuable, and some of them helped us rethink our arguments.

We want to say a few words about how we made the corrections. In the corrected manuscript, we used the terms "rev1" and "rev2" for the corrections related to the comments of reviewers 1 and 2, respectively. As you may expect, the line numbering you used in your comments changed because of the space occupied by the corrections.

**2 **Response to Reviewer 1**

**Comment 1**

Lines 66-67: Why did the authors artificially increase the electron mass? This should only apply to models that need to resolve the plasma frequency, which is what places the constraint on the simulation time step. If the authors simply wanted to mimic the parameter values used in previous works, they may say so. Otherwise, they should justify this choice.

**Response**

We chose the same simulation parameters as Oppenheim(2008) to compare our results to theirs.

**Comment 2**

Lines 78-79: Can the authors elaborate on "we see dominant wave modes growing distinctly faster"? What makes the fastest-growing wave modes dominant other than the fact that they're growing fastest?

**Response**

We just wanted to point out that there is a clear dominant mode in the linear regime, but we can see now that this was redundant because the dominant mode, by definition, grows faster.

**Comment 3**

Line 79: What is the geometry of E and B? It would help the reader to not have to refer to previous work in order to determine the ExB direction.

**Response**

The electric and magnetic fields are parallel to y and z, respectively. We have added this clarification to the text.

**Comment 4**

Lines 79-83: Have the authors made any attempt to quantify the angle of deflection from ExB? Comparing that angle between regularized and unregularized models, as well as to previous results from PIC simulations, would support the authors' main argument. The presentation would benefit from a figure that shows the 2D wavenumber spectrum corresponding to each stage.

**Response**

The main goal of this paper is to show that the five-moment fluid system can remain stable and capture several qualitative features of the Farley-Buneman instabilities. We think showing the small but visible tilt in the density structures from Figure 1 is enough for a qualitative argument. A systematic quantitative analysis of this and other metrics is a topic we are currently working on, but it will be published in the future.

**Comment 5**

Line 82: The note about "a slight turning of the waves in the direction consistent with linear theory" shouldn't apply to the turbulent regime. Did the authors mean to associate this with the linear (or possibly mixing) regime?

**Response**

Even though the analytical estimations of the wave-turning effect have been derived for the linear regime, we are assuming that the preferential direction criteria should still hold approximately as the system goes through saturation. Nevertheless, we agree that the text is ambiguous, so we have modified it to restrict the comment for the linear regime.

**Comment 6**

Have the authors considered changing the resolution while maintaining the simulation domain size, to see if the dominant wavelength changes?

**Response**

As mentioned before, a more systematic evaluation of diagnostics and numerics is part of a current project. And although we have done this analysis and haven't seen a significant change, we decided not to elaborate on this because we were focusing on the qualitative aspects of the results. One of the main contributions of this work is to challenge the standard assumption that simulating Farley-Buneman instabilities in the fluid domain would quickly produce small-scale structures growing faster than the larger ones. In our simulations, we see a dominant mode, so the dependence between growth rate and wavenumber is not monotonically increasing as the standard linear theory predicts. Even though the exact growth rate could change with the resolution of the system, we are assuming that it will maintain the same general behavior.

**Fluid Models Capturing Farley–Buneman Instabilities**

Response to Reviewer 2

**March 22, 2023**

Enrique Rojas1, K.J. Burns2, D.L. Hysell1 elr96@cornell.edu

1Department of Earth and Atmospheric Sciences, Cornell University 2Department of Mathematics, Massachusetts Institute of Technology

**Contents**

| 1 | General Comments to Reviewer 2 |  |
|---|--------------------------------|--|
| 2 | Response to Reviewer 2         |  |
|   | Comment 1                      |  |
|   | Comment 2                      |  |
|   | Comment 3                      |  |
|   | Comment 4                      |  |
|   | Comment 5                      |  |
|   | Comment 6                      |  |
|   | Comment 7                      |  |
|   | Comment 8                      |  |

**1 General Comments to Reviewer 2**

First, we want to thank you for your time writing your thorough review. Your comments were very valuable, and some of them helped us rethink our arguments.

We want to say a few words about how we made the corrections. In the corrected manuscript, we used the terms "rev1" and "rev2" for the corrections related to the comments of reviewers 1 and 2, respectively. As you may expect, the line numbering you used in your comments changed because of the space occupied by the corrections.

**2 Response to Reviewer 2**

**Comment 1**

What thermal effects do the authors refer to in the "including thermal effects" statement?

**Response**

We meant to say that the system solved in this work includes an energy equation, therefore, relaxing the isothermal assumption. We now see that "including thermal effects" might mislead the reader into thinking that Hassan did not include a pressure term in the momentum equation. We have modified the text to be more explicit about this.

**Comment 2**

Why do the authors treat the stress tensor divergence as "artificial viscosity"?

**Response**

Given that the precise form of the term was not justified and its similarity to the neutral viscosity term, it is our impression that this term was added as a numerical proxy for Landau damping. Nevertheless, we now think that it would be more appropriate just to mentioned that this was the operator proposed by the author. The text has now been changed accordingly.

**Comment 3**

Did the authors learn about the role that the stress tensor divergence plays as a good sink for the energy sources to stratify the conservation of energy in Hassan's 2015 fluid model as shown in Figure 8 in Hassan E, Hatch DR, Morrison PJ, Horton W., Multiscale equatorial electrojet turbulence: Energy conservation, coupling, and cascades in a baseline 2D fluid model. Journal of Geophysical Research: Space Physics. 2016 Sep;121(9):9127-45?

**Response**

Yes. Even though the viscosity term used in that work is important for weakly-ionized plasmas, it is not crucial for the paper's central argument. We want to show that the fluid equations without any viscosity (either physical or numerical) can avoid a growth rate of  $k^2$ .

**Comment 4**

Could the inclusion of the energy equation in the fluid model remove the need for the divergence of stress tensor closure?

**Response**

As we mentioned in line 93, we observed a similar behavior using an isothermal system,

namely, no  $k^2$  growth and a dominant band of wave numbers similar to the PIC simulation. Nevertheless, you are right to assume that the energy equation provides an additional source of stabilization to the system. A more comprehensive numerical analysis of these terms is a topic of current work.

**Comment 5**

Including the thermal effects in studying the Farley-Bunemann instability in the aurora region is important, but how important is the inclusion of the thermal effects in simulations at the equatorial electrojet?

**Response**

Even though we don't expect thermal effects to dominate in the equatorial electrojet, they could be relevant if the experiments are precise enough to measure the details of its dynamics. Nevertheless, this paper aimed not to argue that the presented model is more realistic but to compare its results with PIC simulations.

**Comment 6**

Using a small simulation box size couldn't justify the authors' claim about the scalability of the fluid model as running simulations in larger boxes shouldn't use large computational resources.

**Response**

As we mentioned in the conclusions, our claim of computational efficiency relies on the assumption that fluid simulations are computationally cheaper than kinetic simulations if the parameters are similar and kinetic effects are not required.

**Comment 7**

Using a small simulation box limits the destabilization of modes of longer wavelength by energy cascading which might the comparison of the presented results in this manuscript and both the PIC model in Oppenheim et al. 2008 and Hassan et al. 2015.

**Response**

Even though the energy cascading to longer wavelengths is an issue, the paper's primary goal is to show that a simple fluid system avoids the  $k^2$  growth rate. Our argument for scalability is mainly related to the computational complexity of the kinetic solvers. Nevertheless, we have added one sentence to comment on the long wavelength issue.

**Comment 8**

Figure 2: Would the authors explain the difference in the ion and electron heating in the unregularized and regularized cases while having the root-mean-square of the density and electric field almost the same?

**Response**

This might be related to the regularized system restricting the velocity magnitude, making the collisional heating term in the energy equation smaller.